# Purification of Methylsulfonylmethane from Mixtures Containing Salt by Conventional Electrodialysis

**DOI:** 10.3390/membranes10020023

**Published:** 2020-02-01

**Authors:** Xinlai Wei, Yaoming Wang, Haiyang Yan, Ke Wu, Tongwen Xu

**Affiliations:** 1CAS Key Laboratory of Soft Matter Chemistry, Collaborative Innovation Center of Chemistry for Energy Materials, School of Chemistry and Material Science, University of Science and Technology of China, Hefei 230026, China; xinlai@mail.ustc.edu.cn (X.W.); oceanyan@mail.ustc.edu.cn (H.Y.); 2Collaborative Innovation Center for Environmental Pollution Precaution and Ecological Rehabilitation of Anhui, School of Biology, Food and Environment Engineering, Hefei University, Hefei 230601, China; wuke@hfuu.edu.cn

**Keywords:** conventional electrodialysis, methylsulfonylmethane, separation, current efficiency

## Abstract

Methylsulfonylmethane (MSM) is one of the main sources of sulfur for living bodies, but it is hard to obtain as a pure compound. Conventional electrodialysis (CED) is a mature technology that can be used for the separation and purification of biochemical products. In this study, the purification of MSM from mixtures containing salt was performed by CED. The effects of operating conditions such as operation voltage drop, feed MSM concentration, and electrolyte salt concentration on the separation performances were investigated. The results showed that the current efficiency reached 74.0%, and the energy consumption could be 12.3 Wh·L^−1^. As for the recovery rate and desalination rate, the highest recovery rate could be 97.4%, and the desalination rate was 98.5%. Based on process energy consumption calculation, the total cost of the whole process was estimated at only 2.34 $·t^−1^. Thus, CED is highly efficient and cost-effective for the separation and purification of MSM.

## 1. Introduction

Methylsulfonylmethane (MSM), molecular formula (CH_3_)_2_SO_2_, is an important organosulfur compound with antioxidant and anti-inflammatory properties [1]. It is also known by several other names including methyl sulfone, dimethyl sulfone, and DMSO_2_ [2,3]. MSM is widely used in organic synthesis, as an agriculture chemical, and as a high-temperature solvent for both inorganic and organic substances [2,4]. It is one main source of sulfur that is used for the synthesis of methionine, cysteine, sulfur-containing tissue, protein, and peptide in the human body and animals [5]. Therefore, MSM is also commonly used as a dietary supplement that can improve various metabolic diseases [6].

Currently, the most common method for MSM preparation is accomplished by the oxidation of dimethyl sulfide or DMSO using chemical oxidation and electrochemical methods [7]. Chemical oxidation methods include the nitric acid method, hydrogen peroxide method, ozone method, and so on. Electrochemical oxidation methods include the PbO_2_ electrode oxidation method and the graphite electrode oxidation method [1]. Regardless of the preparation method applied, the raw MSM product usually coexists with a certain amount of sodium nitrate salt. The product purity is important for the application scope. High-purity MSM has high-added value for pharmaceutical use, but a low-purity product can only be used as animal feed. It is a demanding task for enterprisers to prepare MSM with low salt. 

In general, the conventional separation and purification of MSM procedures include decolorization by active carbon, demineralization by ion exchange, and then vacuum-drying crystallization. Even though these purifying procedures can obtain pure MSM, these conventional preparation routes have many drawbacks. Firstly, these conventional preparation routes contain complex procedures including decolorization, ion exchange, recrystallization, evaporation, drying, etc. Secondly, the desalination procedure by ion exchange and recrystallization has a high consumption of solvent and chemicals, resulting in a large amount of saline wastewater. It is well known that the regeneration of ion exchange resin has a high consumption of acids and bases, leading to the disposal of saline wastewater. Thirdly, the conventional purifying methods have high energy consumption and high operating cost. Therefore, it is necessary to explore an environmentally friendly, cost-effective, and high-efficiency desalination technology to obtain a high-purity MSM product with low salt. 

Conventional electrodialysis (CED) is a mature separation technology that is widely used in brine desalination, bioproduct demineralization, and salt concentration for zero-liquid discharge (ZLD) [8,9]. By taking advantage of the perm-selectivity of ion exchange membranes under direct-current (DC) electric fields, charged ions migrate from the diluate chamber to the concentrate chamber, while the uncharged components are retained in the diluate chamber. As a consequence, charged and uncharged components are separated from each other [9,10]. Unlike other separation techniques, CED does not suffer from the use of hazardous solvents and chemicals. When it comes to the separation of inorganic salt from the bioproducts, the robustness and competitivity of CED in comparison to other demineralization techniques become more obvious [11]. CED makes it very easy to separate inorganic salt from the relatively low-dissociation bioproducts with little energy consumption. In fact, there were numerous studies on the electrodialytic purification of organic/amino acids [9,11,12,13,14,15,16]. However, to the best of our knowledge, the separation of pure MSM from reaction mixtures containing salt is not reported in the literature. Therefore, the main objectives of this study are to test the feasibility of CED for the purification of MSM from mixtures containing salt, and to investigate the effect of operation voltage drop, feed MSM concentration, and electrolyte salt concentration on the separation performance in terms of the energy consumption and process economy.

## 2. Materials and Methods 

### 2.1. Materials

The anion exchange membrane (AEM; CJ-MA-2) and cation exchange membrane (CEM; CJ-MC-2) were supplied by Hefei Chemjoy Polymer Materials Co. Ltd (Hefei, China). MSM raw material was supplied by Hengjie Chemical co. Ltd (Chongqing, China). The content of salt in MSM dry raw material was 3.14 wt % (calculated on the basis of solid quality of NaNO_3_). Table 1 shows the main properties of the anion exchange membrane and cation exchange membrane used in the experiments.

### 2.2. CED Set-Up

The membrane stack for CED was supplied by Hefei Chemjoy Polymer Materials Co. Ltd (Hefei, China). The components of the membrane stack and experimental procedure are shown in Figure 1. The diluate chamber and concentrate chamber were separated by an ion exchange membrane. The membrane stack was connected to a DC power supply (WYL1703, Hangzhou Siling Electrical Instrument Co. Ltd, Hangzhou, China) through two titanium electrodes coated with ruthenium. The AEM and CEM were separated by a 0.8-mm-thick silica gel partition net. It should be noted that the ED experiments were performed in potentiostatic mode. Constant voltages of 5 V, 10 V, 20 V, and 30 V were applied to the ED stack. When DC current passes through the stack, cations and anions are transported across the CEM and the AEM into the concentrate chamber. The active area of each piece of membrane was 189 cm^2^. Two repeating units were used for the CED stack. A 0.3 mol·L^−1^ Na_2_SO_4_ solution was used in the electrode compartment. Conductivity in the diluate chamber was measured by a conductivity meter. Different concentrations of MSM solution and tap water were added to the diluate chamber and the concentrate chamber, respectively. The volume of each chamber was 400 mL. The flow rate was about 4 L·min^−1^ per cell. At the start of each experiment, the solution was circulated in each chamber for 30 min. When the conductivities of diluate chamber solution were below 500 μs·cm^−1^, experiments were stopped. All experiments were carried out at room temperature.

### 2.3. Analytical Methods

The concentrations of MSM were determined based on total organic carbon (TOC) (Vario TOC select, Elementar, Germany). At first, the standard line of MSM was measured and plotted. The concentration of MSM could be calculated using the standard line, and the regression coefficient was 0.995. 

### 2.4. Calculation of Current Efficiency, Conversion Rate, and Energy Consumption

The desalination rate of MSM (*D*, %) was calculated using Equation (1) [8].
(1)$$D=\frac{\sigma_{t}}{\sigma_{0}}\times 100\%,$$
where *σ*_0_ and *σ_t_* (ms·cm^−1^) are the conductivities in the diluate chamber at time 0 and *t*, respectively.

The recovery rate (*R*, %) of the MSM product was calculated using Equation (2).
(2)$$R=\frac{C_{t\;}}{C_{0}}\times 100\%,$$
where *C*_0_ and *C_t_* are the concentrations of MSM in the diluate chamber at time 0 and *t*, respectively. The energy consumption (*E*, Wh·L^−1^ or kWh·t^−1^) of the CED process was calculated according to Equation (3) [18].
(3)$$E=\frac{U\int_{0}^{t}\mathrm{Idt}}{V},\;$$
where *U* (V) is the voltage drop of CED stack, including the electrode compartments, *I* (A) is the current of the stack, *V* (L) is the volume of the feed solution (it should be noted that the volume change of each chamber was neglected during the experiment (*V* = 0.4 L)), and *t* is the time.

The current efficiency (*η*, %) was calculated using Equation (4) [18].
(4)$$\eta =\frac{Z\left({C^{\prime }}_{t}-{C^{\prime }}_{0}\right)VF}{N\int_{0}^{t}\mathrm{Idt}}\times 100\%,\;$$
where ${C^{\prime }}_{0}$ and ${C^{\prime }}_{t}$ are the molar concentrations of NaNO_3_ (mol·L^−1^) at time *0* and *t* (s), respectively, *Z* is charge number of ions (*Z* = 1 for NaNO_3_), *V* (L) is the volume of the salt compartment, *F* is Faraday’s constant (96,485 C·mol^−1^), *I* (A) is the current, and *N* is the number of repeating units (*N* = 2) in the CED stack.

## 3. Results and Discussion

### 3.1. Effect of Operating Voltage 

Voltage or current is the predominant factor affecting the desalination performance of the electrodialysis process. Here, the CED was operated in potentiostatic mode, and the operating voltages were chosen in the range of 5–30 V. The concentration of MSM was 17.9%. The applied voltage was directly read from the power supply, including the electrode compartment voltage. The contribution of the electrode compartment decreases and the energy consumption becomes smaller if we increase the number of membrane pairs. Figure 2 shows the evolution of conductivities in the diluate chamber under different operating voltage. It can be seen that the conductivities in the diluate chamber (or feed chamber) decreased with the lapse of time. This indicated the successful operation of the CED stack, and the ionic salt was removed from the solutions. Because the raw MSM was composed of MSM and NaNO_3_, the latter represented the main contribution to conductivity. With the depletion of dissociated NaNO_3_ salt ions, the feed solution conductivity inevitably decreases. Meanwhile, it was found that the time required for the desalination experiment decreased with an increase in operating voltage. This is reasonable because the driving force for CED is the current field. A higher current denotes a stronger driving force. However, the conductivity evolution curve for the operating voltage of 20 V was identical to that for the operating voltage of 30 V. This was due to the maximum limitation of power supply (the maximum current was 5.44 A); there was no obvious difference for the current applied at the CED stack between the operating voltages of 20 V and 30 V. The slopes of the conductivity curve decreased with time elapsed. The conductivities in the feed solutions decreased rapidly in the early period of experiments and then slowly descended in the later period of experiments.

Figure 3 shows the effect of voltage drop on the desalination rate and recovery rate of the MSM solution. The desalination rates increased with an increase in voltage drop, and the desalination rates were all higher than 88.1% for different voltage drops. This means that the salt in the MSM mixture was almost completely removed by CED. All salt was transferred from the diluate chamber to the concentrate chamber. In addition, the recovery rates slightly increased with an increase in voltage drop, and the recovery rates were all higher than 87.3%. In fact, in a CED experiment, the loss of un-dissociated compounds along with the dissociated salt is a prevalent issue. In this case, the loss of MSM during the experiment may be due to two reasons. One is molecular diffusion and the other one is electro-migration [8]. The former is caused by the concentration gradient of MSM between the diluate and concentrate chamber; the free MSM molecules diffuse through the membranes into the concentrate chamber. Electro-migration occurs when the MSM is charged after a clustering reaction with metal ions. Buncel et al. [19] reported the clustering reaction between MSM and alkali-metal ions, such as lithium ions. In that case, MSM combines with alkali-metal ions such as Na^+^ ions, and then migrates into the concentrate chamber, as shown in Scheme 1. This electro-migration is more pronounced with an increase in voltage drop. The slight increasing trend of recovery rate with an increase in voltage drop suggests that the diffusion was the overwhelming reason for the loss of MSM during the electrodialysis experiment. Nevertheless, both high desalination rates and recovery rates verified the feasibility of CED for the selective removal of undesired salt.

Figure 4 shows the effect of operating voltage drop on energy consumption and current efficiency in the electrodialysis desalination process. It is indicated that the energy consumption increased with increasing voltage drop. A higher voltage drop results in more energy consumption needed to overcome the electrical resistance [20,21,22]. As a consequence, energy consumption increases accordingly. On the contrary, the current efficiency decreases with increasing operating voltage. The undesired phenomena in the CED process, such as water splitting and co-ions migration [21,22], are more serious with a higher voltage drop, which decreases the current efficiency to a large extent. When the operating voltage was 30 V, the energy consumption was about 5.5 times and the current efficiency was 77% that of 5 V. The desalination rate was slow at low operating voltage. Energy consumption was lower, and the current efficiency was higher at low operating voltage compared to high operating voltage. 

### 3.2. Effect of Feed MSM Concentration

The influence of feed MSM concentration on the desalination process was investigated. When the concentrations of MSM feed solution were 5%, 10%,20%, and 25%, the NaNO_3_ concentrations were 0.17%, 0.35%, 0.70%, and 0.83%, respectively. The operating voltage was 10 V. Figure 5 shows the evolutions of conductivities in the diluate chamber under different feed MSM concentration. It can be seen that the conductivities of diluate chamber dramatically decreased within the first minutes of operation and then decreased slowly. A higher initial concentration of MSM led to a higher salt content in the feed solution. As a consequence, there was a rapid decrease in the conductivity for a high feed MSM concentration due to a low resistance of stack at high salt concentration. The conductivities of the diluate chamber were also below 500 µS·cm^−1^ after the experiments. It can be understood that the experiments took a little more time for a high feed MSM concentration than for a low feed MSM concentration. The salt concentrations and initial conductivities increased with the increase in raw MSM concentration. To reach the same desalination performance, more time was required for the experiment at higher feed MSM concentration. 

Figure 6 shows the desalination rates and recovery rates after 30 min for different feed MSM concentrations. Under an optimal operating voltage drop of 10 V, the desalination rates were all higher than 97.6% for the four feed MSM concentrations. When the feed MSM concentration was 5%, the desalination rate and recovery rate were 98.5% and 97.5%, respectively. The recovery rates decreased with an increase in feed MSM concentration. For example, the recovery rate was 87.5% at a feed MSM concentration of 25%. This was much lower than the 97.4% at a feed MSM concentration of 5%. The MSM concentration gradient between the diluate and concentrate chamber increased with an increase in feed MSM concentration. Therefore, more MSM molecules diffused across the membranes into the concentrate chamber. This caused the loss of MSM in the diluate chamber and a decrease in recovery rate. To obtain a high recovery rate, the CED is recommended to be operated with a low feed MSM concentration solution. 

Figure 7 indicates the energy consumption and current efficiency for different concentrations of MSM. It shows that the energy consumption increased with an increase in MSM concentration. A higher concentration of MSM led to a higher energy consumption. In fact, the total amount of MSM increased with increasing feed MSM concentration. It is expected that the energy consumption would increase, as the energy consumption refers to kilowatts per cubic meters. The current efficiency followed a similar trend and increased with increasing feed MSM concentration. A high concentration of MSM was beneficial for reducing the current resistance, thus improving the current efficiency. 

### 3.3. Effect of the Electrolyte Salt Concentration

In the CED process, the salt in the feed solution continuously migrates into the concentrate compartment, resulting in an increasing concentration of NaNO_3_ in the concentrate chamber. If the salt concentration in the concentrate chamber is very high, there is back diffusion of salt into the diluate chamber owing to molecular diffusion. The direction of back diffusion of salt is inverse to the electrolytic transport. When the back diffusion overwhelms the electrodialytic transport, the desalination of salt cannot be achieved. In that case, replacement of the electrolyte solution is required. Additionally, water osmosis owing to the osmotic pressure difference between the diluate and concentrate chamber is very serious when the salt concentration in the concentrate chamber is very high. Therefore, it is necessary to investigate the effect of electrolyte concentration on the separation performance. In our case, the concentrations of NaNO_3_ in the concentrate chamber were 3%, 6%, 9%, and 12%, respectively. Figure 8 indicates the evolution of conductivities for different kinds of electrolyte concentration in the concentrate chamber. Within the first 5 min, the decreasing rate of conductivity was very similar for all kinds of electrolyte concentration. After that, the degradation rate of conductivity was different; this may have been due to the back diffusion rate of salt from the concentrate chamber to the diluate chamber being different for different salt concentrations. At 20 min, the conductivity in the feed solution was below 500 µs·cm^−1^ for the initial salt concentration of 3%. The conductivities of all samples changed little after an experiment time of 20 min. At that period, the back diffusion of salt from the concentrate chamber to the diluate chamber was in equilibrium with the electrodialytic transport of salt from the diluate chamber to the concentrate chamber. Therefore, to obtain qualified MSM products, a lower initial salt concentration results in a better MSM product quality.

Figure 9 shows the energy consumption and current efficiency at the experimental time of 30 min for different electrolyte concentrations. It is indicated that the desalination rate decreased with an increase in NaNO_3_ concentration and a highest desalination rate of 97.8% was obtained. It is easily understood that a higher concentration in the concentrate chamber led to more salt needing to be transported from the diluate chamber to the concentration chamber. Therefore, the desalination rate decreased with an increase in NaNO_3_ concentration. In addition, the recovery rates were almost the same for all electrolyte concentrations. This implies that the diffused amount of MSM molecules into the concentrate chamber was identical for different electrolyte salt concentrations. The electrolyte salt concentration did not affect the transport of MSM across the ion exchange membranes. 

Figure 10 indicates the energy consumption and current efficiency for different electrolyte salt concentrations. The energy consumption increased with an increase in NaNO_3_ concentration. A higher salt concentration in the concentrate chamber could decrease the overall resistance of the CED stack. However, the increase in energy consumption could be due to the back diffusion of salt to the diluate chamber. Then, the back-diffused salt must be transported back to the concentrate by CED, which results in higher energy consumption. Thus, energy consumption would increase accordingly. The current efficiency decreased with an increase in electrolyte salt concentration. A possible reason is that the back diffusion of salt from the concentrate chamber to the diluate chamber was enhanced with an increase in salt concentration. 

### 3.4. Process Economy

To provide some references for the industrialization of this CED technology, the process economy of this process was estimated [23,24], and the results are shown in Table 2. It should be noted that the process was estimated under the operating voltage of 10 V and MSM of 20%. After an electrodialytic experiment time of 30 min, the conductivity of diluate chamber was below 500 µs·cm^−1^, which meets the requirement for salt content. The total process cost was estimated to be 2.34 $·t^−1^. Figure 11 provides a comparison of the digital image products before and after CED treatment. After desalination using CED, MSM is a white crystalline solid at room temperature and pressure. Considering the high price of MSM, CED is highly efficient and cost-effective for the separation and purification of MSM.

## 4. Conclusions

The purification of MSM was carried out using a laboratory-scale CED device. The effects of operation voltage, feed MSM concentration, and electrolyte salt concentration on voltage drop, desalination rate, recovery rate, energy consumption, and current efficiency were investigated. The process economy of CED was also estimated. It was found that the recovery rates slightly increased with an increase in voltage drop, suggesting that molecular osmosis was the overwhelming reason for the loss of MSM during the electrodialysis experiment. The recovery rates decreased with increasing feed MSM concentration. The MSM concentration gradient between the diluate and concentrate chamber increased with increasing feed MSM concentration. To obtain a high recovery rate, the CED is recommended to be operated with a low feed MSM concentration. The recovery rate was not significantly affected by electrolyte salt concentration, but the current efficiency decreased with an increase in electrolyte salt concentration. In the CED process, the current efficiency reached 74.0%, and energy consumption could be 12.3 Wh·L^−1^. As for the recovery rate and desalination rate, the highest recovery rate could be 97.4%, and the desalination rates were all higher than 98.5%. Based on process energy consumption calculation, the total cost of the whole process was only 2.34 $·t^−1^. Thus, CED is cost-effective and highly efficient for the separation and purification of MSM.

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
