# Peer review of "Purification of Methylsulfonylmethane from Mixtures Containing Salt by Conventional Electrodialysis"

_membranes, 2020, doi:10.3390/membranes10020023_

Round 1

Reviewer 1 Report

The submitted manuscript describes the desalination of a mixture containing Methylsulfonylmethane (MSM) by ion exchange membrane electrodialysis. The experiments carried out here are interesting and related to the aim of the study. However, there are some serious issues which affect the manuscript quality seriously. The main issue is the English which is very poor. Thus, I would suggest the rejection of the article in current form. However, the author can address the comment and resubmit the manuscript.

 Abstract: Eng: current efficiency could up to 74.0%

21: So CED is very environment friendly; this is not true as sometime the byproduct of the ED is a problem for environment which must be further processed

29: English: MSM is widely and large scale applied as a medium in agriculture chemical

31: English: It is one of the main sources of sulfur that is in the  synthesis of methionine,

40: English: The purity of MSM  is a key parameter that determine

41: English: But low purity product can only use as animal feed.

42: English: To obtain MSM with low salt  is a demanding process for the enterprisers.

44: English: In general, the conventional separation and purification of MSM procedures including decolored by active carbon, demineralized by ion exchange, and then recrystallized the solvent, dried by vacuum.

46: English: these conventional  preparation route has many drawbacks.

 47: English Firstly, this conventional preparation route contains complex  procedures including de-coloration, ion exchange, recrystallization, evaporation, dry, etc.

55: English: that has widely applications in brine desalination

60: Different  from the other separation techniques, CED does not suffer from the generation of large amounts of 61 waste: this is not true as usually there is large amount of concentrate directed to sewage

65: English: reports about the electrodialytic purify

67: English: has never reported in the literature

2.1 Materials

The functional group used in each kind of the membrane must be mentioned

The method applied for determining the membrane resistance must be explained

2.2

Please pay attention that in electrodialysis the dilution does not take place ( it is not simple dilution for example as with water), hence,  using the term dilute chamber is not appropriate. The common term is diluate chamber which has been used in a lot of reference books in electrodialysis.

Why did you titanium coated with ruthenium electrodes in electrodialysis.

It is very important to explain which regime was used? Potentiostatic or the galvanostatic, explain which potential or current was applied.

86:English: cation transported pass through the  CEM into the concentrate chamber, anion transported across the AEM into the concentrate chamber.

2.4

Where U (V) is the voltage drop of CED stack; Please specify which potential do you mean?  If it is potential of membranes stack only or the electrodes compartment are included as well.

3.1

128: English: the conductivity in the feed  solution will inevitably decrease.

146: English: the experiment is possibly ascribed to two reasons.

152:English In that case, MSM will combine with these alkali-metal ions such as Li+ , Na+ , K+ 152 , etc and transport into the concentrate chamber

164: English: It is indicated that energy consumption is increased

164: It is indicated that energy consumption is increased with an increase in voltage drop. The higher the voltage drop, the more energy used to overcome the electrical resistance. As consequence, energy consumption will increase accordingly.

I do not understand this statement. The resistance depends on the membranes and diluate and concentrate solution. So the resistance is constant. However, the energy consumption at higher operating voltage is higher because the desalination is faster and therefore the resistance of the diluate becomes higher in the main period of the CED.

3.2

177: English: The influence of feed MSM concentration on the desalination process were investigated.

178: English: The  concentration of MSM raw material solution were 5%, 10%,20% and 25%, respectively.

179 : English: The operating 179 voltage were 10V.

199: English: The recovery rates are 199 decreased

201: English: the concentration-gradient of MSM between the dilute and concentrate chamber is accelerated.

210: English: In fact, the total amount of MSM 211 is increase

213: English: Current efficiency follows a similar trend and increase

256: The higher the concentration of NaNO3, the more energy is used to overcome the electrical resistance:

I disagree with this statement. Higher concentration of the salt in concentrate decreases the overall resistance of the stack. However, the increase in energy consumption could be due to the back diffusion of salt to diluate. Then the back diffused salt must be transported back to the concentrate by CED which results in higher energy consumption.

Conclusion

277:Pure MSM was prepared by a laboratory-scale CED device: Better to say that the purification of MSM was carried out by a laboratory-scale CED device

277: The effect of operation voltage, feed 278 MSM concentration, electrolyte salt concentration on the separation performance were investigated. You must specify the effects of the parameters on what were investigated.

279: English: The energy consumption, current efficiency, process economy was evaluated:

285: English: The recovery rate is not significant affected.

Author Response

Response. We appreciate the reviewer’s constructive comments on this manuscript. For the poor language problem, we have proofread the manuscript carefully to correct the grammatical, spelling and typo errors. In addition, we also asked several colleagues who are skilled speakers of English language to improve the English. We deeply appreciate the efforts reviewer’s efforts in pointing out the language errors line-by-line.

Q1. Abstract: Eng: current efficiency could up to 74.0%.

A1. This sentence has revised as “current efficiency reached 74%”.

Q2. 21: So CED is very environment friendly; this is not true as sometime the byproduct of the ED is a problem for environment which must be further processed.

A2. We agree with the reviewer’s comments that sometime the byproduct of the ED is a problem for environment which must be further processed. In the abstract, the claim about “environment friendly of ED” has deleted.

Q3. 29: English: MSM is widely and large scale applied as a medium in agriculture chemical

A3. This sentence has revised as “MSM is widely used in organic synthesis, as an agriculture chemical and as a high-temperature solvent for both inorganic and organic substances”.

Q4. 31: English: It is one of the main sources of sulfur that is in the  synthesis of methionine,

A4. This phrase has revised as “It is one main source of sulfur element that is used for the synthesis of methionine”.

Q5. 40: English: The purity of MSM  is a key parameter that determine

A5. This phrase has revised as “The product purity is important for the application scope”.

Q6. 41: English: But low purity product can only use as animal feed.

A6. This phrase is revised as “but low purity product can be used only as animal feed”.

Q7. 42: English: To obtain MSM with low salt  is a demanding process for the enterprisers.

A7. This phrase is revised as “It is a demanding task for the enterprisers to prepare MSM with low salt”.

Q8. 44: English: In general, the conventional separation and purification of MSM procedures including decolored by active carbon, demineralized by ion exchange, and then recrystallized the solvent, dried by vacuum.

A8. This sentence is revised as “In general, the conventional separation and purification of MSM procedures are including decolorization by active carbon, demineralization by ion exchange and then vacuum-drying crystallization”.

Q9. 46: English: these conventional  preparation route has many drawbacks.

A9. This phrase is revised as “these conventional preparation routes have many drawbacks”.

 Q10. 47: English Firstly, this conventional preparation route contains complex  procedures including de-coloration, ion exchange, recrystallization, evaporation, dry, etc.

A10. This sentence has revised as “Firstly, these conventional preparation routes contain complex procedures including decolorization, ion exchange, recrystallization, evaporation, drying and etc”.

Q11. 55: English: that has widely applications in brine desalination

A11. This phrase has revised as “that has widely used in brine desalination”.

Q12. 60: Different  from the other separation techniques, CED does not suffer from the generation of large amounts of 61 waste: this is not true as usually there is large amount of concentrate directed to sewage

A12. Thanks for the reviewer’s comments. Now we have deleted the statement about CED does not suffer from the generation of large amounts waste.

Q13. 65: English: reports about the electrodialytic purify

A13. This phrase has revised as “there were numerous studies of the electrodialytic purification”.

Q14. 67: English: has never reported in the literature

A14. This phrase has revised as “has never been reported in the literature”.

Q15. 2.1 Materials The functional group used in each kind of the membrane must be mentioned. The method applied for determining the membrane resistance must be explained.

A15. Thanks for the comments. The functional groups of membranes used in the experiments were sulfonic acid groups for CJMC-2 and quaternary ammonium groups for CJMA-2. The main properties of membranes used for the experiments were collected from the product brochure provided by the manufacturers. The membrane resistances were tested with 0.5 mol/L NaCl aqueous solution at room temperature. Now this information has added in Table 1 as required.

Q16. 2.2. Please pay attention that in electrodialysis the dilution does not take place ( it is not simple dilution for example as with water), hence,  using the term dilute chamber is not appropriate. The common term is diluate chamber which has been used in a lot of reference books in electrodialysis.

A16. Thanks for the reviewer’s comments. According to the reviewer’s suggestion, the term “dilute chamber” has revised as “diluate chamber” throughout the manuscript.

Q17. Why did you titanium coated with ruthenium electrodes in electrodialysis.

A17. Titanium coated with ruthenium electrode is relative cheap and stable compared to other electrodes coated with noble metal such as platinum. These kind electrodes are feasible to work in an aqueous solution with high concentration of chloride or sulfate ions, and are feasible to work with alternated frequently pole-reversing current. For these reasons, titanium coated with ruthenium electrodes were used in this experiment.

Q18. It is very important to explain which regime was used? Potentiostatic or the galvanostatic, explain which potential or current was applied.

A18. The ED experiments were performed at a potentiostatic mode. Constant voltages of 5V, 10V, 20V and 30V were applied to the ED stack. This information has added in the experiment part (Line 86-87).

Q19. 86:English: cation transported pass through the  CEM into the concentrate chamber, anion transported across the AEM into the concentrate chamber.

A19. This phrase has revised as “cations and anions are transported across the CEM and the AEM into the concentrate chamber”.

Q20. 2.4 Where U (V) is the voltage drop of CED stack; Please specify which potential do you mean?  If it is potential of membranes stack only or the electrodes compartment are included as well.

A20. Here the U (V) is the potential of the whole ED stack, the voltage drops of electrode compartments are also included. A special note of the voltage drop has added in the text.

Q21. 3.1 128: English: the conductivity in the feed  solution will inevitably decrease.

A21. This phrase has revised as “the feed solution conductivity will decrease inevitably”.

Q22. 146: English: the experiment is possibly ascribed to two reasons.

A22. This phrase has revised as “the loss of MSM during the experiment may be due to two reasons”.

Q23. 152:English In that case, MSM will combine with these alkali-metal ions such as Li+ , Na+ , K+ 152 , etc and transport into the concentrate chamber

A23. This sentence has revised as “In that case, MSM will be combining with these alkali-metal ions such as Li+, Na+ and K+, and then be migrating into the concentrate chamber”.

Q24. 164: English: It is indicated that energy consumption is increased

A24. This sentence has revised as “It is indicated that the energy consumption increases with increasing voltage drop”.

Q25. 164: It is indicated that energy consumption is increased with an increase in voltage drop. The higher the voltage drop, the more energy used to overcome the electrical resistance. As consequence, energy consumption will increase accordingly.

I do not understand this statement. The resistance depends on the membranes and diluate and concentrate solution. So the resistance is constant. However, the energy consumption at higher operating voltage is higher because the desalination is faster and therefore the resistance of the diluate becomes higher in the main period of the CED.

A25. In the experiment, the energy consumption increases with an increase in voltage drop. It is actually indeed that the resistance depends on the membranes and diluate and concentrate solution. If the current efficiency is the same at different voltage, the energy consumption should be identical for different voltage because the amount of salt is the same. But now the energy consumption increases with increasing voltage. Therefore, a greater part of electrical energy is consumed to overcome the electrical resistance as the voltage increases. This explanation is consistence with the other studies that reported in the literature (AICHE J., 2006, 52, 393-401; Separation and Purification Technology, 2018, 194, 416-424). These references were added into the revised manuscript.

Q26. 3.2 177: English: The influence of feed MSM concentration on the desalination process were investigated.

A26. Now “were” is corrected as “was”.

Q27. 178: English: The concentration of MSM raw material solution were 5%, 10%,20% and 25%, respectively.

A27. This sentence has revised as “The concentrations of MSM feed solution were 5%, 10%, 20% and 25%, respectively”.

Q28. 179 : English: The operating 179 voltage were 10V.

A28. Now “were” is corrected as “was”.

Q29. 199: English: The recovery rates are 199 decreased

A29. This sentence has revised as “The recovery rates decrease with an increase of feed MSM concentration”.

Q30. 201: English: the concentration-gradient of MSM between the dilute and concentrate chamber is accelerated.

A30. This sentence has revised as “The MSM concentration gradient between the diluate and concentrate chamber increases with an increase of feed MSM concentration”.

Q31. 210: English: In fact, the total amount of MSM 211 is increase

A31. This sentence has revised as “In fact, the total amount of MSM increases with increasing feed MSM concentration”.

Q32. 213: English: Current efficiency follows a similar trend and increase

A32. This sentence has revised as “Current efficiency follows a similar trend and increases with increasing feed MSM concentration”.

Q33. 256: The higher the concentration of NaNO3, the more energy is used to overcome the electrical resistance:

I disagree with this statement. Higher concentration of the salt in concentrate decreases the overall resistance of the stack. However, the increase in energy consumption could be due to the back diffusion of salt to diluate. Then the back diffused salt must be transported back to the concentrate by CED which results in higher energy consumption.

A33. Thanks for the reviewer’s kind comment. We agree with the reviewer’s viewpoint that the increase in energy consumption could be due to the back diffusion of salt to diluate. This explanation has added in the manuscript (Line 269-273).

Q34. Conclusion. 277:Pure MSM was prepared by a laboratory-scale CED device: Better to say that the purification of MSM was carried out by a laboratory-scale CED device

A34. Done as suggested.

Q35. 277: The effect of operation voltage, feed 278 MSM concentration, electrolyte salt concentration on the separation performance were investigated. You must specify the effects of the parameters on what were investigated.

A35. Thanks for the comments. This sentence has revised as “The effect of operation voltage, feed MSM concentration and electrolyte salt concentration on voltage drop, desalination rate, recovery rate, energy consumption and current efficiency were investigated”.

Q36. 279: English: The energy consumption, current efficiency, process economy was evaluated:

A36. This sentence has revised as “The process economy of CED was also estimated”.

Q37. 285: English: The recovery rate is not significant affected.

A37. This phrase has revised as “The recovery rate is not significantly affected”.

Reviewer 2 Report

The manuscript contains a useful information and therefore it can be published after some corrections, I’ve listed below in order of appearance in the text.

Title: „Preparation of pure methylsulfonylmethane by conventional electrodialysis” – „preparation”?? it is a purification of MSM solutions.

„Dimethyl Dulfone” – ?

„certain amount of sodium nitrate salts” – „salts”?

„the main objectives of this study are to test the feasibility of CED for the preparation of pure MSM,” – „preparation of pure MSM” –> purification of MSM solutions.

“MSM raw material was supplied by Hengjie Chemical co. Ltd. The content of salt in MSM raw material is 0.58 wt% (calculated on the solid quality of NaNO3).” – were other impurities in that material?

Table 1: Water Uptake – what was the counterion?

Resistance, Transfer number – what was the electrolyte solution, its concentration, temperature?

„The active area of the membranes was 189 cm2 per cell.” – what was the area of single membrane?

How many compartment pairs – 2, as in Fig.1?

How was measured the applied voltage? Was it read from the power supply?

If yes what about the electrode voltage drop? If we increase the number of the membrane pairs then the contribution of that component will decrease and the energy consumption will be smaller. Any comments will be useful.

Eq.(1) lambda is symbol for the molar conductivity not for specific conductance. Please, use IUPAC terminology.

Eq.(2) – it would be better to express R as a ratio of the number of moles; the concentration can change due to the volume flux (electroosmosis), osmosis).

What about volume changes during the ED process? – they are not reported – it should be completed!

Eq.(4) – in the denominator there is It not integral. Is I = const? According to e.g. Fig.2 U is constant, thus I should depend on time.

the ion’s absolute valence – charge number of ion.

„.. the driven force for a CED is the current filed.” - ??

„..the loss of MSM within the experiment is possibly ascribed to two reasons. One is the molecular diffusion osmosis and the other one is electro-osmosis [8].” – please, use a correct terminology: osmosis is a movement of solvent and diffusion is  a movement of solute, both caused by concentration difference (or gradient), electro-osmosis is a movement of solvent caused by migrating ions in the externally applied electric field.

„the MSM is charged by a hydrated ion cluster reaction with metal ions” – „MSM is charged by a reaction with metal ions (Scheme 1)”?

„MSM will combine with these alkali-metal ions such as Li+, Na+, K+, etc” – were Li+, K+ found in the raw material? If yes what was their concentration?

„The undesired phenomena in CED process such as water splitting, co-ions migration and concentration-gradient diffusion [17], are more serious in higher voltage drop” – why co-ions migration and, especially, diffusion should be „more serious” in higher voltage drop? Please, explain.

“concentration-gradient diffusion” – just „diffusion”.

Figs.5, 6, 7: as NaNO3 is the main factor influencing the changes of conductivities and other quantities the concentration of NaNO3 should also be given.

In the raw material the concentration ratio of NaNO3 and MSM is 0.58/99.42 = 0.0058.

In Fig.5 25% MSM corresponds to 0.145% NaNO3 solution. The conductivity of that solution at time zero is higher than 7 mS/cm. From literature data the conductivity of 0.5% NaNO3 solution is only 5.4 mS/cm at 20 C. For 0.145% NaNO3 it would be only ca. 1.6 mS/cm.  How to explain such a high discrepancy?

„With an increase of feed MSM concentration, the concentration-gradient of MSM between the dilute and concentrate chamber is accelerated.” – „concentration-gradient” -> „concentration difference”, „accelerated”??

Fig. 7 Energy consumption and current efficiency under different concentration of MSM” – here the additional axis for the concentration of NaNO3 should be given.

„Fig. 7 indicates the energy consumption and current efficiency under different concentration of MSM.” – what was the time of ED?

„3.3. Effect of the electrolyte salt concentration” – why 3-12% of NaNO3 are used whereas in the raw product there is only 0.58% ? Please, justify.

Table 7 – formulae for the calculation of some quantities should be given. E.g. treatment capacity is 7.68 T/a. But if I take feed solution 0.4 L, time 30 min, 1 year – 8760 h then

8760 h*0.4 L / 0.5 h = 7.0 T/a not 7.68 T/a (if T denotes 1000 kg but in that case it should be t, not T).

Author Response

Response. We appreciate the reviewers very much for your positive and constructive comments and suggestions on our manuscript. We have studied your comments carefully and have made revisions. For the language, we have proofread the manuscript carefully and also asked colleagues of skilled English speaker to improve the language.

Q1. Title: Preparation of pure methylsulfonylmethane by conventional electrodialysis” – „preparation”?? it is a purification of MSM solutions.

A1. Thanks for the reviewer’s comment. We have revised the title as “Purification of methylsulfonylmethane from mixtures containing salt by conventional electrodialysis”

Q2. Dimethyl Dulfone” – ?

A2. Sorry there is a typo mistake. Now “Dimethyl Dulfone” has corrected as “Dimethyl Sulfone”.

Q3. certain amount of sodium nitrate salts” – „salts”?

A3. Thanks. Now “Salts” has revised as “salt” throughout the manuscript.

Q4. the main objectives of this study are to test the feasibility of CED for the preparation of pure MSM,” – „preparation of pure MSM” –> purification of MSM solutions.

A4. We agree with the reviewer that this experiment is actually a purification process. This phrase has revised as “purification of MSM from mixtures containing salt”.

Q5. “MSM raw material was supplied by Hengjie Chemical co. Ltd. The content of salt in MSM raw material is 0.58 wt% (calculated on the solid quality of NaNO3).” – were other impurities in that material?

A5. We are sorry there is an error for the salt concentration. The content of salt in MSM dry raw material is about 3.14 wt% (calculated on the solid quality of NaNO3). Here the 0.58% is the salt concentration in the aqueous solution when MSM concentration is 17.9%. There are no other impurities in the raw material because an initial pretreatment has been conducted by the enterpriser to get this raw material.

Q6. Table 1: Water Uptake – what was the counterion?

Resistance, Transfer number – what was the electrolyte solution, its concentration, temperature?

A6. The data in Table 1were collected from the product brochure provided by the manufacturers. The counterions of CJMC-2 and CJMA-2 are chloride ions and sodium ions when the water uptake is tested. The resistances were tested with 0.5 mol/L NaCl solution at room temperature. The transport number was tested with 0.1 and 0.2 mol/L KCl solution at room temperature. Now this information has added in Table 1.

Q7. The active area of the membranes was 189 cm2 per cell.” – what was the area of single membrane?

A7. This sentence has revised as “The active area of each piece of membranes was 189 cm2”.

Q8. How many compartment pairs – 2, as in Fig.1?

A8. Yes. Now the repeating unit of CED has added in the manuscript. (Line 90)

Q9. How was measured the applied voltage? Was it read from the power supply?

If yes what about the electrode voltage drop? If we increase the number of the membrane pairs then the contribution of that component will decrease and the energy consumption will be smaller. Any comments will be useful.

A9. The applied voltage is directly read from the power supply. It is actually indeed as the reviewer said that the contribution of electrode compartment will decrease and the energy consumption will be smaller if we increase the number of the membrane pairs. In this experiment, the CED is operated at a potentiostatic mode. The current is changed with the evolution of experiment, so the contribution of electrode resistance to whole stack resistance is also changed. The electrode voltage along with the experiment is not on-line determined. But for this CED stack, the relation between the electrode voltage and current is known, which follows an empirical equation, V=2.8771 I0.2349 (R² = 0.9979; V-Voltage; I-Current) (as shown in the following figure). For voltage drop of 5V, the initial current is 1.35 A at time 0 min. The electrode voltage is about 3.08V, accounting for 61% for the whole stack voltage. But at the end of experiment (the experiment time of 60 min), the current is about 0.05A. The electrode voltage is about 1.42 V, accounting for 28% for the whole stack voltage. A notice about the applied voltage includes the electrode compartment voltage has added in the text (Line 127-130).

Figure The dependence of electrode resistance with current

Q10. Eq.(1) lambda is symbol for the molar conductivity not for specific conductance. Please, use IUPAC terminology.

A10. Done as suggested. The symbol of conductivity has changed as sigma.

Q11. Eq.(2) – it would be better to express R as a ratio of the number of moles; the concentration can change due to the volume flux (electroosmosis), osmosis).

A11. In this laboratory-scale experiment, the volume change of each compartment is neglected. The recovery express as a ratio of the number of moles is the same as a ratio of ion concentration.

Q12. What about volume changes during the ED process? – they are not reported – it should be completed!

A12. In this laboratory-scale experiment, the volume change of each compartment is neglected. This description has added in the text. (Line 114-115)

Q13. Eq.(4) – in the denominator there is It not integral. Is I = const? According to e.g. Fig.2 U is constant, thus I should depend on time.

A13. Thanks for point out this mistake, we have corrected this error.

Q14. the ion’s absolute valence – charge number of ion.

A14. Done as suggested.

Q15. the driven force for a CED is the current filed.” - ??

A15. Thanks. We have corrected this typo error.

Q16. the loss of MSM within the experiment is possibly ascribed to two reasons. One is the molecular diffusion osmosis and the other one is electro-osmosis [8].” – please, use a correct terminology: osmosis is a movement of solvent and diffusion is a movement of solute, both caused by concentration difference (or gradient), electro-osmosis is a movement of solvent caused by migrating ions in the externally applied electric field.

A16. Thanks for the comment. We have revised this terminology as “One is the molecular diffusion and the other one is electro-osmosis”.

Q17. the MSM is charged by a hydrated ion cluster reaction with metal ions” – „MSM is charged by a reaction with metal ions (Scheme 1)”? MSM will combine with these alkali-metal ions such as Li+, Na+, K+, etc” – were Li+, K+ found in the raw material? If yes what was their concentration?

A17. The inorganic salt of the raw material is NaNO3, the sodium ions concentration is about 9.5 g/L. The Li+ and K+ were not found in the raw material. Therefore, we have replaced lithium with sodium in Scheme 1.

Q18. The undesired phenomena in CED process such as water splitting, co-ions migration and concentration-gradient diffusion [17], are more serious in higher voltage drop” – why co-ions migration and, especially, diffusion should be „more serious” in higher voltage drop? Please, explain.

A18. At higher voltage drop, the current is higher. The selectivity of ion exchange membranes decreases with an increase in current density (AICHE J. 2006, 52, (1), 393-401). In that case, the co-ions migration should be more serious with a decrease of selectivity of membranes. This reference has added in the text. The diffusion should be not related to voltage drop, so this sentence was rephrased as “The undesired phenomena in CED process such as water splitting and co-ions migration, are more serious in higher voltage drop”.

Q19. “concentration-gradient diffusion” – just „diffusion”.

A19. Done as suggested.

Q20. Figs.5, 6, 7: as NaNO3 is the main factor influencing the changes of conductivities and other quantities the concentration of NaNO3 should also be given.

A20. When the MSM solution concentrations are 5%, 10%, 20% and 25%, the NaNO3 concentrations are 0.17%, 0.35%, 0.70% and 0.83%, respectively. This information has added in the caption of Figures 5-7.

Q21. In the raw material the concentration ratio of NaNO3 and MSM is 0.58/99.42 = 0.0058.

In Fig.5 25% MSM corresponds to 0.145% NaNO3 solution. The conductivity of that solution at time zero is higher than 7 mS/cm. From literature data the conductivity of 0.5% NaNO3 solution is only 5.4 mS/cm at 20 C. For 0.145% NaNO3 it would be only ca. 1.6 mS/cm. How to explain such a high discrepancy?

A22. We are sorry this is an error for the salt concentration in the dry raw material. The concentration of salt in the dry raw material is 3.14%. Here the 0.58% is the salt concentration in the aqueous solution when MSM concentration is 17.9%.

Q23. With an increase of feed MSM concentration, the concentration-gradient of MSM between the dilute and concentrate chamber is accelerated.” – „concentration-gradient” -> „concentration difference”, „accelerated”??

A23. We have revised this sentence as “The MSM concentration gradient between the diluate and concentrate chamber increases with an increase of feed MSM concentration”.

Q24. Fig. 7 Energy consumption and current efficiency under different concentration of MSM” – here the additional axis for the concentration of NaNO3 should be given.

A24. When the MSM solution concentrations are 5%, 10%, 20% and 25%, the NaNO3 concentrations are 0.17%, 0.35%, 0.70% and 0.83%, respectively. This note has added in the caption of Fig. 7. An additional axis for the concentration of NaNO3 was also added in Fig. 7 as suggested.

Q25. Fig. 7 indicates the energy consumption and current efficiency under different concentration of MSM.” – what was the time of ED?

A25. The CED experimental time is in the range of 31-37 minutes when the MSM concentration is 5-25%.

Q26. 3.3. Effect of the electrolyte salt concentration” – why 3-12% of NaNO3 are used whereas in the raw product there is only 0.58% ? Please, justify.

A26. As described in the text, the salt in feed solution will continuously migrate into the concentrate compartment, resulting in increasing concentration of NaNO3 in the concentrate chamber. If the salt concentration in the concentrate chamber is very high, there will be back-diffusion of salt into the diluate chamber owing to molecular diffusion. Therefore, we need to replace the electrolyte salt when NaNO3 concentration in the concentrate chamber is very high. The 3-12% of NaNO3 is about 5-20 times of the feed salt concentration. These concentration times are the limitation ranges for a CED concentration process. So, 3-12% of NaNO3 is used.

Q27. Table 7 – formulae for the calculation of some quantities should be given. E.g. treatment capacity is 7.68 T/a. But if I take feed solution 0.4 L, time 30 min, 1 year – 8760 h then

8760 h*0.4 L / 0.5 h = 7.0 T/a not 7.68 T/a (if T denotes 1000 kg but in that case it should be t, not T).

A27. Here T denotes ton. We have corrected it.

Reviewer 3 Report

This work is based on purification of methylsulfonylmethane (MSM) using conventional electrodialysis which is more energy efficient and cost effective compared to other methods that contain several complex steps and cause losses. In this work, the authors have separated pure MSM from sodium nitrate impurity. Also, they have carried out a thorough study of desalination rate, effect of voltage drop and energy efficiency in this report. This manuscript is scientifically sound but suffers from poor English, consisting of both spelling and grammatical errors. I recommend publication after the authors significantly improve the writing and check manuscript for typos. 

Author Response

Response. Thanks for the reviewer’s positive comment on this manuscript. We have proofread the manuscript carefully and also asked several colleagues of skilled English speaker to improve the language. We sincerely appreciate your valuable comments and suggestions, which helped us to improve the quality of the manuscript.

Round 2

Reviewer 1 Report

The revised version of manuscript is improved in quality and I am satisfied with the corrections made therein. Therefore, Ii would recommend for the publication in Membranes.

Kind regards

Author Response

We deeply appreciate the reviewer’s constructive comments on this manuscript. Thank you very much.

Reviewer 2 Report

There are stiil some points to be improved:

Table 1:

“The transport number was tested with 0.1 and 0.2 mol/L” – “tested”?  better “determined”.

Water uptake strongly depends on the kind of counterions – it would be good to give the counterion (if this info is available).

Eq.(3) – U = const, so it should be moved outside the integral to avoid any doubts.

Eq.(4) – N = const, so it should be moved outside the integral to avoid any doubts.

„One is molecular diffusion and the other one is electro-osmosis [8]. The former is caused by the concentration gradient of MSM between the diluate and concentrate chamber; the free MSM molecules will diffuse through the membranes into the concentrate chamber. The latter electro-osmosis is happening when the MSM is charged by a hydrated ion cluster reacted with metal ions.” – please, use a correct terminology: electro-osmosis refers to a movement of solvent, not solute; here we have migration of the charged MSM in the externally applied electric field.

„the MSM is charged by a hydrated ion cluster reacted with metal ions.” – this formulation seems to me a little bit strange. „a hydrated ion cluster reacted with metal ions” - ??

„Fig. 7 Energy consumption and current efficiency …” – please, give here the time of ED.

Still the formulae for the calculation of some quantities in Table 7 are lacking. E.g.: could you, please, explain how the value of the treatment capacity (7.68 t/a) was obtained? If I take the volume of feed solution 0.4 L, time 30 min, 1 year – 8760 h then:

8760 h*0.4 L / 0.5 h = 7.01 t/a, not 7.68 t/a.

In Table 3 “T” should be replaced with “t” not only in the treatment capacity.

English should be improved, e.g.:

“The direct of back diffusion…” – direction,

Author Response

Response to reviewer #2

Reviewer #2’s Comment:

There are stiil some points to be improved:

Response. Thanks for the reviewer’s comments for help us improving the quality of the manuscript.

Q1. Table 1: “The transport number was tested with 0.1 and 0.2 mol/L” – “tested”?  better “determined”.

A1. Done as suggested.

Q2. Water uptake strongly depends on the kind of counterions – it would be good to give the counterion (if this info is available).

A2. The counterions of cation- and anion- exchange membrane for water uptake determination were Na+ and Cl- ions, respectively. This information has added in the note of Table 1.

Q3 . Eq.(3) – U = const, so it should be moved outside the integral to avoid any doubts.

Eq.(4) – N = const, so it should be moved outside the integral to avoid any doubts.

A3. Thanks for the comment. U and N have moved outside the integral as requested.

Q4.One is molecular diffusion and the other one is electro-osmosis [8]. The former is caused by the concentration gradient of MSM between the diluate and concentrate chamber; the free MSM molecules will diffuse through the membranes into the concentrate chamber. The latter electro-osmosis is happening when the MSM is charged by a hydrated ion cluster reacted with metal ions.” – please, use a correct terminology: electro-osmosis refers to a movement of solvent, not solute; here we have migration of the charged MSM in the externally applied electric field.

A4. Thanks for the comments. Now we have corrected the “electro-osmosis” as “electro-migration”.

Q5. the MSM is charged by a hydrated ion cluster reacted with metal ions.” – this formulation seems to me a little bit strange. „a hydrated ion cluster reacted with metal ions” - ??

A5. We have revised this phrase as “The latter electro-migration is happening when the MSM is charged after a clustering reaction with metal ions”

Q6. Fig. 7 Energy consumption and current efficiency …” – please, give here the time of ED.

A6. The energy consumption was calculated when the conductivity in the diluate chamber is below 500 µs·cm-1. The time of ED was 10mins, 12.5 mins, 20 mins and 20 mins for feed MSM concentration of 5%, 10%, 20% and 25%, respectively.

Q7. Still the formulae for the calculation of some quantities in Table 7 are lacking. E.g.: could you, please, explain how the value of the treatment capacity (7.68 t/a) was obtained? If I take the volume of feed solution 0.4 L, time 30 min, 1 year – 8760 h then:

8760 h*0.4 L / 0.5 h = 7.01 t/a, not 7.68 t/a.

A7. Thanks very much for point out this mistake. The treatment capacity is 7.01t/a; the total fixed cost is 1.48$·t-1 and total process cost is 2.34 $·t-1. Some remarks was added in this table to explain how the value is calculated.

Q8. In Table 3 “T” should be replaced with “t” not only in the treatment capacity.

A8. Done as suggested.

Q9. English should be improved, e.g.:

“The direct of back diffusion…” – direction,

A9. We have corrected it and proofread the manuscript again to improve the language. Thanks again.